# Cortical Granule Distribution and Expression Pattern of Genes Regulating Cellular Component Size, Morphogenesis, and Potential to Differentiation are Related to Oocyte Developmental Competence and Maturational Capacity In Vivo and In Vitro

**DOI:** 10.3390/genes11070815

**Published:** 2020-07-17

**Authors:** Magdalena Kulus, Wiesława Kranc, Michal Jeseta, Patrycja Sujka-Kordowska, Aneta Konwerska, Sylwia Ciesiółka, Piotr Celichowski, Lisa Moncrieff, Ievgeniia Kocherova, Małgorzata Józkowiak, Jakub Kulus, Maria Wieczorkiewicz, Hanna Piotrowska-Kempisty, Mariusz T. Skowroński, Dorota Bukowska, Marie Machatkova, Sarka Hanulakova, Paul Mozdziak, Jędrzej M. Jaśkowski, Bartosz Kempisty, Paweł Antosik

**Affiliations:** 1Department of Veterinary Surgery, Institute of Veterinary Medicine, Nicolaus Copernicus University in Torun, 87-100 Torun, Poland; magdalena.kulus@umk.pl (M.K.); pantosik@umk.pl (P.A.); 2Department of Anatomy, Poznan University of Medical Sciences, 60-781 Poznan, Poland; wkranc@ump.edu.pl (W.K.); ikocherova@ump.edu.pl (I.K.); 3Department of Obstetrics and Gynecology, University Hospital and Masaryk University, 602 00 Brno, Czech Republic; jeseta@gmail.com; 4Department of Veterinary Sciences, Czech University of Life Sciences in Prague, 165 00 Prague, Czech Republic; 5Department of Histology and Embryology, Poznan University of Medical Sciences, 60-781 Poznan, Poland; psujka@ump.edu.pl (P.S.-K.); akonwer@ump.edu.pl (A.K.); sciesiolka@ump.edu.pl (S.C.); pcelichowski@ump.edu.pl (P.C.); l.moncrieff.16@abdn.ac.uk (L.M.); 6Department of Anatomy and Histology, University of Zielona Gora, 65-046 Zielona Gora, Poland; 7School of Medicine, Medical Sciences and Nutrition, University of Aberdeen, Aberdeen AB25 2ZD, UK; 8Department of Toxicology, Poznan University of Medical Sciences, 60-631 Poznan, Poland; malgorzata.jozkowiak@gmail.com (M.J.); hpiotrow@ump.edu.pl (H.P.-K.); 9Department of Diagnostics and Clinical Sciences, Institute of Veterinary Medicine, Nicolaus Copernicus University in Torun, 87-100 Torun, Poland; jakub.kulus@umk.pl (J.K.); dbukowska@umk.pl (D.B.); jedrzej.jaskowski@gmail.com (J.M.J.); 10Department of Basic and Preclinical Sciences, Institute of Veterinary Medicine, Nicolaus Copernicus University in Torun, 87-100 Torun, Poland; maria.wieczorkiewicz@umk.pl (M.W.); skowron@umk.pl (M.T.S.); 11Veterinary Research Institute, 621 00 Brno, Czech Republic; machatkova@vri.cz (M.M.); sarka.hanulakova@mendelu.cz (S.H.); 12Prestage Department of Poultry Science, North Carolina State University, Raleigh, NC 27695, USA; pemozdzi@ncsu.edu

**Keywords:** pig, oocyte maturation, microarray, cortical granule, molecular markers

## Abstract

Polyspermia is an adverse phenomenon during mammalian fertilization when more than one sperm fuses with a single oocyte. The egg cell is prepared to prevent polyspermia by, among other ways, producing cortical granules (CGs), which are specialized intracellular structures containing enzymes that aim to harden the zona pellucida and block the fusion of subsequent sperm. This work focused on exploring the expression profile of genes that may be associated with cortical reactions, and evaluated the distribution of CGs in immature oocytes and the peripheral density of CGs in mature oocytes. Oocytes were isolated and then processed for in vitro maturation (IVM). Transcriptomic analysis of genes belonging to five ontological groups has been conducted. Six genes showed increased expression after IVM (*ARHGEF2*, *MAP1B*, *CXCL12*, *FN1*, *DAB2*, and *SOX9*), while the majority of genes decreased expression after IVM. Using CG distribution analysis in immature oocytes, movement towards the cortical zone of the oocyte during meiotic competence acquisition was observed. CGs peripheral density decreased with the rise in meiotic competence during the IVM process. The current results reveal important new insights into the in vitro maturation of oocytes. Our results may serve as a basis for further studies to investigate the cortical reaction of oocytes.

## 1. Introduction

Knowledge of molecular mechanisms that protect oocytes against polyspermia is important both in basic research and in the future of human medicine, as well as to improve the efficiency of in vitro production (IVP) in animals. It is known that there are two mechanisms that prevent more than one sperm from entering the oocyte [1]. The first barrier works at the level of the plasma membrane of the oocyte, which is depolarized after sperm fusion and excretion of the egg’s sperm receptor Folr4 (folate receptor 4, also known as Juno) [2]. The second mechanism under consideration is the “cortical reaction”, which consists of exocytosis of cortical granules (CGs) containing enzymes (e.g., ovastacin [3]) that harden the zona pellucida, which reduces the affinity to connect with subsequent sperm [4]. The cortical reaction itself is a well-known phenomenon, but its molecular background is not fully explained. So far, there have been reports related to the transport of CGs from the cytoplasmic center to the periphery of the cell [5,6], docking to the plasma membrane [6] or resulting in exocytosis of CGs [7,8,9], but not fully illustrating these processes. Therefore, it is very significant to know the expression profile of the genes involved in the cortical reaction, which, in turn, will expand the possibilities of detailed research. 

It is generally accepted that frequent abnormal development of in-vitro-produced porcine embryos and early embryonic mortality are often caused by polyspermic penetration. Previous studies described the relationship of higher occurrence of polyspermic penetration with lower meiotic competence, which is characteristic of oocytes of prepubertal gilts [10,11]. Machatkova et al. stated that meiotic and developmental competence is an ability of oocytes to undertake resumption and successful termination of meiosis and is acquired gradually with follicle growth and completed during maturation. This implies that satisfactory levels of meiotic and developmental competence can be achieved only after the onset of the oestrous cycle [12]. 

Due to the requirements for large numbers of oocytes, prepubertal gilts represent the prevailing source of experimental porcine oocytes. Oocytes of prepubertal gilts before the onset of oestrus do not reach the optimal developmental competence, which results in higher polyspermy rate and significantly lower blastocyst yield than in adult sows, despite there being no differences in nuclear maturation [13,14]. The majority of juvenile oocytes from prepubertal gilts are smaller than 3 mm in diameter, and these were proved to be cytoplasmatically deficient, and their ability to finish maturation may be limited [15]. 

Knowledge of the mechanism of oocyte development during folliculogenesis and its effect on developmental potential of the oocyte is an essential prerequisite for improvement of in vitro fertilization (IVF) and in vitro production (IVP) systems. Low monospermic penetration rate and high rate of polyspermy are still the main unsolved problems in porcine embryo production [16]. Although in vitro maturation (IVM) porcine oocytes are successfully cultured up to the blastocyst stage, a high polyspermy rate, often reaching more than 50%, is still an obstacle despite the fact that the oocytes have their own mechanisms for blocking polyspermic penetration [16]. The major role in prevention of polyspermy is played by cortical granules. CGs are secretory vesicles present only in female germ cells. In mammals, they range in size from 0.2 to 0.6 µm in diameter [17] and are formed continuously from the early stages of oocyte growth. During follicular development and maturation, they translocate from throughout the ooplasm to the cortex up until the time of ovulation. Migration of CGs depends on the functioning cytoskeleton and is considered to be a criterion of successful cytoplasmic maturation, which is characterized, among several factors, by redistribution of organelles [18]. Migration of CGs is a preparatory phase for triggering a cortical reaction, which is a calcium-dependent exocytotic process started by sperm–oocyte fusion [19]. This binding activates a range of signaling pathways leading to release of the cortical granules’ content to the perivitelline space. It initiates structural and biochemical modifications of zona pellucida to form the impermeable barrier [20].

The function of CGs was long believed to start and finish by exocytosis of their contents at the moment of oocyte penetration. However, in some recent studies, authors reported that some CGs undergo exocytosis prior to fertilization, and some cortical granules components function beyond the time of fertilization in regulation of embryonic cleavage and preimplantation development [17]. 

The process of the oocyte acquiring maturity is highly advanced and requires the involvement of many genes that directly or indirectly influence the achievement of MII stage. The molecular background during the reorganization of intracellular structures in the maturing oocyte has not been well known. Further research is needed to establish the thorough mechanisms regulating these processes.

The objective of the study was to further understand the expression of genes related to morphogenesis before and after in vitro maturation of swine oocytes to better define gene pathways involved in maturation. Furthermore, the aim was to better understand the distribution and density of CGs in porcine oocytes in regard to meiotic competency for IVM porcine oocytes.

## 2. Materials and Methods 

All chemicals used in the present study were purchased from Sigma-Aldrich Chemicals Co. (Prague, Czech Republic) unless otherwise stated.

### 2.1. Animals, Oocyte Collection, and Selection

Ovaries from adult cycling sows were obtained from an experimental slaughterhouse and transported to the laboratory within 2 h at 31–33 °C. The average age of the animals was 40 months and they were derived from crossbred Landrace X Czech Large White. The ovaries of each donor were evaluated and only those with proper morphologic status (presence of ovarian follicles of different sizes) were used. Ovaries were assessed according to morphology of follicles and presence of corpora lutea.

Oocytes with higher meiotic competence from medium follicles (MF)—5–9 mm and with lower meiotic competence from follicles smaller than 5 mm (SF)—were collected from ovaries by follicle aspiration and cutting of the ovarian cortex, respectively. Oocytes from larger follicles (≥10 mm) were not included in the study. All healthy cumulus–oocyte complexes with dark, evenly granulated ooplasm and at least two cumulus layers were used for the experiments. 

The Brilliant Cresyl Blue (BCB) staining assay was performed for assessment of porcine oocytes’ quality and maturity and for selection of oocytes for later tests. The enzyme glucose-6-phosphate converts the BCB stain from blue to colourless. In oocytes that completed the growth, activity of the enzyme decreases, and the stain cannot be reduced, resulting in blue oocytes (BCB+). Oocytes were rinsed twice with modified Dulbecco PBS (DPBS) (Sigma-Aldrich, St. Louis, MO, USA) supplemented with 50 μg/mL streptomycin, 50 IU/mL penicillin (Sigma-Aldrich, St. Louis, MO, USA), 0.4% BSA (w/v), 0.34 mM pyruvate, and 5.5 mM glucose (DPBSm). They were then treated with 13 μM BCB (Sigma-Aldrich, St. Louis, MO, USA) diluted in DPBSm at 38.5 °C and 5% CO_2_ for 90 min. Then, the oocytes were transferred to DPBSm and washed twice. During the washing procedure, oocytes were examined under inverted microscope and classified as blue stained (BCB+) or colorless (BCB−). Only BCB+ oocytes were used in molecular analyses (“Before IVM” group) or IVM followed by second BCB test and molecular analyses (“After IVM” group).

### 2.2. Oocytes Maturation and Examination

Isolated cumulus–oocyte complexes that were not used for examination at germinal vesicle (GV) stage were matured in groups of 25–30 in 500 µL of M199 medium (with Earle’s salts), supplemented with 0.20 mM sodium pyruvate, 0.57 mM cysteamine, 50 IU mL^−1^ penicillin, 50 µg mL^−1^ streptomycin, 10% BFS (bovine fetal serum), and gonadotropins (P.G.600 15 IU mL^−1^, Intervet, Holland) in a four-well multidish (Nunc, Intermed, Denmark) for 44 h at 39 °C in a humidified atmosphere of 5% CO_2_. The mean maturation rate was 85%. Only oocytes with verified MII stage represented matured oocytes in the experiment. Oocytes in germinal vesicle stage and mature oocytes were manually denuded of cumulus cells in IVM culture medium with 0.1% (w/v) hyaluronidase (Sigma Aldrich). After washing, the oocytes were fixed in 3.7% paraformaldehyde for 60 min at room temperature. They were washed in PBS and mounted on glass slides, avoiding oocyte compression, using Vectashield medium (Vector Lab, Burlingame, CA, USA) containing 1 μM of DNA dye (SYTOX Orange, Invitrogen; Carlsbad, CA, USA) specific for dyeing of chromatin. The slides were stored below 0 °C until examination. The oocytes were examined with the use of a laser scanning confocal microscope (Leica TCS SP2 AOBS; Leica, Heidelberg, Germany) equipped with argon and helium–neon lasers. The 532 nm excitation band and 550–590 nm detector were used for detection of chromatin. The 40× Leica HCX PL APO CS objective, pinhole, offsets, gain, and Acousto-Optical Beam Splitter (AOBS) were adapted.

### 2.3. Cortical Granule Staining and Evaluation of Their Distribution 

The methods for staining CGs were based on those reported by Yoshida et al. [21] and Wang et al. [22] with a few modifications. Immature and in-vitro-matured oocytes were denuded of cumulus cells by vortex. Experimental groups of oocytes were washed three times in PBS containing 0.4% BSA and then fixed with 3.7% paraformaldehyde for 1 h at room temperature, followed by three washes in PBS containing 0.4% BSA for 10 min each. This procedure was followed by treatment with 1% Triton X-100 for 1 h and washing three times in PBS containing 0.4% BSA for 10 min each time. Oocytes were then cultured in 10 μg/mL fluorescein isothiocyanate (FITC)-labelled Peanut Agglutinin (PNA) in PBS for 30 min in a dark box. After staining, the immature oocytes were washed three times in PBS containing 0.4% BSA for 10 min. In-vitro-matured oocytes were washed three times in PBS containing 0.4% BSA for 2 min. Nuclear status of oocytes was visualized by staining of the oocytes using Vectashield medium (Vector Lab, Burlingame, CA, USA) containing 1 μM of DNA dye (SYTOX Orange, Invitrogen). Finally, the oocytes were mounted on slides with teflon anticompression layer and observed under a laser scanning confocal microscope (Leica TCS SP2 AOBS; Leica, Heidelberg, Germany) equipped with an argon ion laser for visualization of cortical granules via excitation of FITC and a helium–neon laser for excitation of Sytox Orange visualizing DNA. The 40× Leica HCX PL APO CS objective, pinhole, offsets, gain, and AOBS were adjusted. These parameters were maintained throughout the whole experiment. The oocytes were scanned in equatorial optical sections and microphotographs were saved and processed using the NIS-ELEMENTS AR 3.0 software (Laboratory Imaging, Prague, Czech Republic). A total of 185 immature oocytes were evaluated for CGs distribution. Two parts in each optical section of oocyte were measured—the central area, which occupied 80%, and the peripheral area, which occupied 20% of the optical section. The movement of CGs was assessed with the index of cortical granules distribution (I_cgd_). The Index of CGs distribution represents the ratio between mean intensity signal in the peripheral part of oocyte and the mean intensity signal in the central part of oocyte. In matured oocytes (*n* = 197) all CGs were localized in the peripheral area. The density of CGs from each optical section, scanned at the apical pole of each oocyte, was counted at three different areas on each optical section. Number of CGs/100 µm^2^ in each oocyte was counted.

Microphotographs of CGs distribution from confocal microscopy were saved and processed using the NIS-ELEMENTS AR 3.0 software (Laboratory Imaging, Prague, Czech Republic). The values of I_cgd_ and density of cortical granules were statistically analyzed by the independent samples *t*-test using STATISTICA CZ, version 10 software (StatSoft, Inc., Prague, Czech Republic). Differences with *p* < 0.05 were considered as statistically significant.

### 2.4. Histological Examination 

For histological purposes, ovaries from five animals were collected. Immediately after collection, samples were fixed in the Bouin’s solution for 48 h. Consequently, ovaries were dehydrated with increasing concentrations of ethyl alcohol and xylen and then embedded in paraffin blocks. Subsequently, they were cut into 3–4 μm thick sections with a semiautomatic rotary microtome (Leica RM 2145, Leica Microsystems, Nussloch, Germany). All ovary sections were stained with a routine hematoxylin and eosin (H&E) staining method following the protocol: deparaffinization and rehydration, staining with H&E, and dehydration. Histological sections were evaluated by light microscope and selected pictures were taken with a use of high-resolution scanning technique and Mirax Midi BF/FL microscope scanner (Carl Zeiss MicroImaging GmbH, Göttingen, Germany).

### 2.5. RNA Isolation

The oocyte samples were divided into two groups. The first group contained pooled oocytes (SF and MF) before IVM, the second group included pooled oocytes (SF and MF) after IVM. Each group was combined into three independent samples representing separate experimental groups. Total RNA was extracted from the samples with RNeasy MinElute Kit Cleanup (Qiagen, Hilden, Germany) and TRI Reagent^®^ (Sigma, St. Louis, MO, USA). The obtained amount of total mRNA was measured using the optical density at 260 nm, and the purity of RNA from the absorption ratio 260/280 nm (above 1.8) (NanoDrop spectrophotometer, Thermo Scientific, ALAB, Waltham, MA, USA). RNA quality was determined with the use of Bioanalyzer 2100 (Agilent Technologies, Inc., Santa Clara, CA, USA). Obtained RNA integrity (RIN) ranged from 8.5 to 10 with an average of 9.2 (Agilent Technologies, Inc., Santa Clara, CA, USA). RNA was diluted to a 100 ng/L concentration with an OD260/OD280 ratio of 1.8/2.0. Each sample was separated and secured for further analysis; only 100 ng was used for microarray assays. The remaining RNA was used for RT-qPCR analysis.

### 2.6. Microarray Expression Analysis and Statistics

All the experiments were carried out in triplicate [23,24]. Total RNA (100 ng) from each pooled sample was subjected to two-round sense cDNA amplification (Ambion^®^ WT Expression Kit, Thermo Fisher Scientific, Inc., Waltham, MA, USA). The cDNA was used for biotin labeling and fragmentation by Affymetrix GeneChip^®^ WT Terminal Labeling and Hybridization (Affymetrix, Thermo Fisher Scientific, Santa Clara, CA, USA). Biotin-labeled fragments of cDNA (5.5 μg) were hybridized to Affymetrix^®^ Porcine Gene 1.1 ST Array Strip (48 °C/20 h). Then, microarrays were washed and stained according to the technical protocol, using Affymetrix GeneAtlas Fluidics Station (Affymetrix, Thermo Fisher Scientific, Santa Clara, CA, USA). The array strips were scanned employing Imaging Station of GeneAtlas System (Affymetrix, Thermo Fisher Scientific, Santa Clara, CA, USA). The preliminary analysis of the scanned chips was performed using Affymetrix GeneAtlasTM Operating Software (version 2.0.0.460; Affymetrix; Thermo Fisher Scientific, Inc. Waltham, MA, USA). Quality of gene expression data was checked according to quality control criteria provided by the software. Obtained CEL files were imported into downstream data analysis software [25,26,27].

All analyses were performed using BioConductor software (open source project; version 3.11; http://www.bioconductor.org/), based on the statistical R (version 3.5.1; www.r-project.org) programming language. For background correction, normalization and summation of raw data, the Robust Multiarray Averaging (RMA) algorithm implemented in “affy” package of BioConductor (open source project; version 3.11; http://www.bioconductor.org/) was applied. Biological annotation was taken from BioConductor “oligo” package where annotated data frame object was merged with normalized data set, leading to a complete gene data table. Statistical significance of analyzed genes was performed by moderated *t*-statistics from the empirical Bayes method. Obtained *p* values were corrected for multiple comparisons using Benjamini and Hochberg’s false discovery rate. The selection of significantly changed gene expression was based on *p* value beneath 0.05 and expression fold higher than |2|. 

Functional annotation clustering of differentially expressed genes was performed using DAVID (Database for Annotation, Visualization and Integrated Discovery) (open source project; version 6.8; DAVID, https://david.ncifcrf.gov/). Gene symbols for up- or downregulated genes from each of the compared groups were loaded to DAVID by “RDAVIDWebService” BioConductor package (open source project; version 3.11; http://www.bioconductor.org/). For further analysis, we chose the enriched GO terms that had at least five genes and *p*-value (Benjamini) lower than 0.05. The enriched GO terms were subjected to the hierarchical clusterization algorithm and presented as heat maps. 

Subsequently, we analyzed the relationship between the genes belonging to chosen GO terms with the GOplot 1.0.2 package (open source project; https://wencke.github.io/) [28]. The GoPlot package calculated the Z-score: the number of upregulated genes minus the number of downregulated genes divided by the square root of the count. This information allowed estimating the change course of each gene ontology term.

Interactions between differentially expressed genes/proteins belonging to an ontology group were investigated by STRING10 software (Search Tool for the Retrieval of Interacting Genes) (open source project; https://string-db.org/). A list of gene names was used as query for interaction prediction. Searching criteria were based on co-occurrences of genes/proteins in scientific texts (text mining), coexpression, and experimentally observed interactions. The results of such analysis generated a gene–protein interaction network where the intensity of the edges reflects the strength of the interaction score. Besides interaction prediction, STRING also allowed us to perform functional enrichments of GO terms based on previously uploaded gene sets.

Finally, the functional interaction between genes that belong to the chosen GO BP terms were investigated by REACTOME FIViz application to the Cytoscape 3.6.0 software (https://cytoscape.org/). The Reactome FIViz (open source project; https://reactome.org/tools/reactome-fiviz) app is designed to find pathways and network patterns related to cancer and other types of diseases. This app accesses the pathways stored in the Reactome database, allowing pathway enrichment analysis for a set of genes, visualizing hit pathways using manually laid-out pathway diagrams directly in Cytoscape 3.6.0 software (open source project; https://cytoscape.org/), and then investigating functional relationships among genes in hit pathways. The app can also access the Reactome Functional Interaction (FI) network, a highly reliable, manually curated, pathway-based protein functional interaction network covering over 60% of human proteins.

### 2.7. Quantitative Analysis of Polymerase Chain Reaction in Real Time (RT-qPCR) Analysis

For RT-qPCR, total RNA isolated earlier from oocytes groups before and/or after IVM was used. RNA samples were resuspended in 20 μL of RNase-free water and stored in a −80 °C freezer. RNA samples were treated with DNase I and reverse transcribed (RT) into cDNA. RT-qPCR was performed in a real-time LightCycler PCR detection system (Roche Diagnostics GmbH, Mannheim, Germany) using SYBRR Green I as a detection dye, with target cDNA quantified using a relative quantification method [29]. The relative abundance of all tested genes transcripts in each sample was normalized to internal standards (PBGD, β-actin, 18S rRNA). For amplification, 2 μL of diluted cDNA was added to 18 μL PCR QuantiTectR SYBRR Green PCR (Master Mix Qiagen GmbH, Hilden, Germany) and primers (Table A1, see the Appendix A). To provide a negative control for subsequent PCR, one RNA sample from each preparation was analyzed without an RT reaction.

## 3. Results

### 3.1. Distribution of CGs in Immature MF and SF Oocytes

The values of the Index of CGs distribution (I_cgd_) in MF and SF oocytes regardless to the size of follicles are presented in Table 1. The difference of the I_cgd_ values between oocytes with higher and lower meiotic competence derived from medium and small follicles, respectively, was statistically significant (*p* < 0.05). I_cgd_ in oocytes with lower meiotic competence (1.48 ± 0.31) shows a tendency of central orientation of CGs in the cytoplasm of oocytes derived from SF (Figure 1). On the opposite, in oocytes with higher meiotic competence the peripheral concentration of CGs was more than two-fold higher (I_cgd_ = 2.22 ± 0.49) than their concentration in the central part, indicating movement of CGs towards the cortical area during the acquisition of meiotic competence.

### 3.2. Peripheral Density of CGs in Mature MF and SF Oocytes 

The comparison of CGs density in the cortical area with regard to size of the follicles from which the oocytes were derived showed statistically significant difference (*p* ˂ 0.05) between these experimental groups (Figure 2). The density of CGs recorded in more meiotically competent oocytes from medium follicles reached a mean value of 59.31 ± 13.2, while in oocytes with lower meiotic competence derived from small follicles, the mean value of CGs density was 73.89 ± 18.7. These results proved that, together with increasing meiotic competence, the numbers of CGs per 100 µm^2^ of cortical area of ooplasm declines during the IVM process. 

### 3.3. Histological Structure of Porcine Ovary and Follicles

Histological analysis allowed us to evaluate the morphological structure of the collected ovaries. Ovarian follicles were observed at all stages of maturation. Clusters of primordial follicles were visible just beneath the tunica albuginea. Primary follicles are characterized by one or many layers of granulosa cells surrounding the oocyte with zona pellucida. In the secondary follicle, formation of antrum was observed between the granulosa cells. In the mature ovarian follicle, the oocyte is located on one of the poles of the follicle, and it is surrounded by cells forming a corona radiata (Figure 3). 

### 3.4. Microarrays Analysis 

Whole transcriptome profiling by Affymetrix microarray allowed us to analyze the gene expression changes in freshly isolated oocytes before the in vitro procedure (“before IVM”) compared to after in vitro maturation (“after IVM”), as pooled samples from SF and MF. Using Affymetrix^®^ Porcine Gene 1.1 ST Array (Affymetrix, Thermo Fisher Scientific, Santa Clara, CA, USA), 12,258 porcine transcripts were analyzed. Genes with fold change higher than |2| and with corrected *p* value lower than 0.05 were considered as differentially expressed. This set of genes consisted of 419 different transcripts. Subsequently, the genes were used for identification of significantly enriched GO BP terms. 

DAVID software was used for extraction of the genes belonging to “cellular component morphogenesis”, “negative regulation of cell differentiation”, “morphogenesis of a branching structure”, “regulation of cellular component size”, and “branching morphogenesis of a tube” Gene Ontology Biological Process terms (GO BP). We found 39 genes belonging to the ontological group GO BP terms described above. From the results obtained during the analysis of microarrays, 33 genes were significantly represented in downregulated, and only six were upregulated after in vitro maturation. These sets of genes were subjected to the hierarchical clusterization procedure and are presented as heat maps (Figure 4). 

Sets of the differentially expressed genes belonging to “cellular component morphogenesis”, “negative regulation of cell differentiation”, “morphogenesis of a branching structure”, “regulation of cellular component size”, and “branching morphogenesis of a tube” GO BP terms with their official gene symbols, fold changes in expression, corrected *p* values, and LogFC are shown in Table 2.

The enrichment of each GO BP term and KEGG (Kyoto Encyclopedia of Genes and Genomes) pathway were calculated as Z-score and are shown on the circle diagram in Figure 5.

Moreover, in the Gene Ontology database, genes that formed one particular GO group can also belong to other different GO term categories. For this reason, we explored the gene intersections between the selected GO BP terms. The relation between those GO BP terms was presented as a circle plot (Figure 6) as well as a heatmap (Figure 7).

A STRING-generated interaction network was created with differentially expressed genes belonging to the “cellular component morphogenesis”, “negative regulation of cell differentiation”, “morphogenesis of a branching structure”, “regulation of cellular component size”, and “branching morphogenesis of a tube” ontology groups. The intensity of the edges reflects the strength of interaction score (Figure 8). Finally, the functional interactions between chosen genes with REACTOME FIViz app to Cytoscape 3.6.0 software (https://cytoscape.org/) are shown in Figure 9.

### 3.5. Microarray Validation—RT-qPCR

The results obtained from the expression microarrays were subjected to quantitative verification using the RT-qPCR technique. The direction of changes in the expression of 37 of 39 genes has been confirmed (Figure 10). Such divergences may be the result of differences in the sensitivity of both methods used (microarray and RT-qPCR). RT-qPCR is a method much more sensitive to changes in gene expression.

## 4. Discussion

The presented research, both at the molecular level of gene expression and distribution of cortical granules, was aimed at identifying processes leading to full maturity by the oocyte. In vitro maturation of oocytes is still difficult, which translates into a decrease in IVF process efficiency.

The transfer of CGs is associated with meiotic maturation of oocytes and is often considered a determinant of maturity. Therefore, it is very important to know the mechanisms regulating the dynamics of CGs. We have evaluated the distribution of CGs in immature oocytes, both from small and medium follicles. There is a tendency towards central CGs orientation in SF oocytes with lower meiotic competence. On the contrary, in MF oocytes, the peripheral CGs concentration was twice as high as the central concentration (Figure 1, Table 1) suggesting the movement of CGs towards the plasma membrane from the central part of the oocyte, which takes place during puberty and meiotic competence acquisition. 

Additionally, the average value of peripheral density of CGs in MF was lower than in SF (Figure 2). Therefore, it can be concluded that with the increase in meiotic competence, the number of CGs per 100 um^2^ of the cortical ooplasm surface decreases during the IVM process. CGs appear to be a dynamic structure, dependent on many factors, and are characterized by a complex distribution during the development of oocytes. The molecular mechanism of formation and releasing of cortical granules are not well known. Many studies emphasize that the fate of the embryo is determined at a very early stage of development, before embryo genome activation (EGA), and even before fertilization. It is also suggested that all mechanical actions on cells are reflected at the molecular level [30].

The microarray study was concentrated on the analysis of genes belonging to “cellular component morphogenesis”, “negative regulation of cell differentiation”, “morphogenesis of a branching structure”, “branching morphogenesis of a tube”, and “regulation of cellular component size” ontological groups by conducting in vitro culture of porcine oocytes in primary culture. The chosen groups refer to the organization of cell structure and its morphogenesis or component size, which may be of key importance for the proper development of oocytes and its usefulness for IVF processes. Analysis of microarrays was performed before and after in vitro maturation of swine oocytes, which enabled us to evaluate molecular markers during IVM. In general, downregulation of the expression of the genes after IVM was observed, except for the group “cellular component morphogenesis”, which, in part, showed upregulation after IVM (Figure 4).

The gene expression profile in in-vitro-maturing porcine oocytes showed an increased expression in relation to the state before IVM. All upregulated genes belonged exclusively to the “cellular component morphogenesis” group, associated with the creation of new cells and their components. The gene that is involved in the organization of cytoskeleton, especially microtubules and actin filaments—ARHGEF2 (Rho/Rac guanine nucleotide exchange factor 2) [31]—is upregulated after IVM. ARHGEF2 regulates the RhoGTP-azes signal pathways, which are important factors in the fundamental cellular processes associated with migration, proliferation, and survival [32]. Zhong et al. [33] described that RhoA is a key element in ooplasmic segregation and spindle rotation, and studies on swine oocytes indicated participation in meiosis regulation and early embryonic development [34]. Similarly, research on in vitro mouse embryos has shown that an impairment of the ARHGEF2 gene associated with the organization of actin may have an impact on embryonic development [35]. However, earlier studies in cows with subclinical endometritis showed a decrease in ARHGEF2 expression, which will probably affect the gene profile of the embryo and its further development [36]. High expression of ARHGEF2 in mature oocytes may, therefore, indicate correct cytoskeleton formation and may be a candidate as an in vitro maturation marker of oocytes.

The second analyzed gene is closely related to the assembly of the microtubules. The MAP1B gene (microtubule associated protein 1B) plays a predominant role in the development of the nervous system and neural flexibility [37], although its participation in other tissues has been shown in kidney podocytes in humans and mice [38], or in ovarian tissues [39]. Reduced expression was shown in studies on swine mammary during later gestation [40]. Upregulation of this gene was demonstrated in swine granulosa cells (GCs) during the analysis of follicular atresia [41] and appeared in studies comparing GCs and theca cells of bovine animals [42]. Budna et al. [43] showed upregulation after porcine oocytes IVM, which was suggested to be related to the reorganization of microtubules in maturing oocytes.

Furthermore, it was demonstrated that the transport of CGs from the central part of the cell to the membrane is actin-dependent [5,6]. It is due to the existence of the actin cytoplasmatic network connected with Rab11a that vesicles can be transferred [44]. Recent studies on mouse oocytes have shown that CGs move along the cytoplasmic actin network in a process regulated by Rab27a (Ras superfamily of GTPases). Additionally, Rab27a mutants showed increased polyspermia [4]. On the other hand, earlier studies have shown that CGs diffuse freely in cytoplasm before meiosis, and binding to microfilaments occurs at the beginning of meiosis, which results in the transfer of CGs to the cortical area [5]. Wessel et al. studied the transport of CGs dependent on cytoskeleton by adding inhibitors such as nocodozoles and colchicine. The addition of these inhibitors did not completely block the transport, but the packing of CGs along the plasma membrane was leaking and incomplete, which resulted in the impossibility of exocytosis. However, addition of microfilament inhibitors (cytochalasin D and latrunculin A) caused an absence of cortical transport. CGs were dispersed in cytoplasm [5], suggesting a key role of the actin network in the transport of CGs, and microtubes were proposed as the motor. Significant increase in expression of ARHGEF2 and MAP1B genes, which are closely related to the structure of cytoskeleton, may indicate the participation of these genes in CGs distribution and, at the same time, indicate the maturity of oocytes. 

A recently published study on oocytes (OCs) from pigs showed no progression of meiosis and poor distribution of CGs after exposure of OCs to BFA (brefeldin A—a beta-lactam antibiotic) [45]. The indisposition of these processes was explained by a clearly defective cytoplasmic system, which consequently weakens the ability to fertilize. Therefore, it is a further proof of the key importance of microfilaments in oocytes maturation and CGs dynamics. Miao et al. described the important role of dynein (cytoskeletal motor proteins) in the meiotic progression of porcine oocytes through the regulation of cytoskeletal dynamics, including the stability of microtubules [46], reflecting the distribution of CGs and ovastacin exocytosis.

The next gene, CXCL12 (C-X-C motif chemokine ligand 12), is an extremely important gene that exhibited an increase in expression after IVM of swine oocytes. CXCL12 is also known as SDF-1 (stromal cell-derived factor) and is a CXC group chemokine with a wide range of effects, including hematopoiesis and embryo development [47]. Due to its receptor in the cell membrane (CXCR4), it has an important role in the mechanism called “cell homing”, which turns out to be an important process in cell therapy with the use of stem cells [47,48,49]. It acts as a chemoattractant, which is suggested by Zuccarello et al. [50] to be important in the mechanism of guiding sperm to the oocyte, because human preovulatory oocytes express SDF-1. The expression of this gene is also documented in GCs and theca cells of cattle [42] and porcine cheeks (downregulation during in vitro culture) [51]. Additionally, it was shown that CXCL12 acts synergistically with VEGF, which induces angiogenesis in human ovarian cancer [52], but also supports angiogenesis after ovulation [53]. Through analyzing literature data and the obtained results, it can be assumed that CXCL12 plays an important role in the maturity of oocytes for fertilization.

In contrast, increased expression of DAB2 (DAB adaptor protein 2) gene was shown. DAB2 is involved in regulation processes such as cellular adhesion [54], which is important in cell culture. The expression of this gene was recorded in granulosa cells and theca cells in bovine, during comparative studies of these two groups of ovarian follicle cells [42,55]. The work of Stefańska et al. [56] observed a high increase in DAB2 expression in oviduct epithelial cells (OECs) in primary culture, which indicated high participation of this gene in morphogenesis. Additionally, this gene is considered important for the functioning of the female reproductive system [57] and plays a role in the TGF-beta pathway as a link between TGF-beta receptors and proteins from the SMAD family [58]. Taking into account the participation of this gene in the processes of reproductive system cells, it can be concluded that it has marker potential, which requires further confirmatory studies.

The FN1 gene (fibronectin) was upregulated in our current studies and is also strongly associated with cellular adhesion processes. Fibronectin is a major component of the extracellular matrix, also involved in the migration of cells associated with the basement membrane from connective tissue [59]. The expression of this gene was also recorded in the aforementioned studies on bovine GCs and theca cells [42] and on swine GCs [25,60], where an increase in expression during primary in vitro culture was observed. Significant expression of this gene in reproductive system tissues may provide insight into its role in ovarian follicle and oocyte morphogenesis.

However, the SOX9 gene (SRY-box transcription factor 9) is associated with the processes related to sex differentiation—its activation of transcription and gonadal development [61,62]. It also has a role in skeletal development and chondrogenesis [63,64,65]. The increase in the expression of this gene may indicate the maturation of gametes from the female gonads. Increased expression of the three previously mentioned genes (FN1, DAB2, and SOX9) was also described in the primary culture of cells obtained from the porcine buccal cheek, which was accompanied by cell proliferation [51]. 

The remainder of the analyzed genes showed downregulation after IVM of swine oocytes in vitro culture. A large group of genes was closely or indirectly related to oocytes, ovarian follicles, or the reproductive system in general. The first gene that belonged only to the “cellular component morphogenesis” of the ontological group was GJA1 (gap junction protein Alpha 1), one of the key factors in oocyte maturation [66,67]. GJA1, also known as CX43, is involved in building gap junction connections (GJCs) between individual cumulus cells (CCs) or between CCs and oocytes. Its expression is particularly manifested in granulosa cells. The protein channels make it possible to transfer small molecules, acquire competence through the oocyte, and resume its meiosis [68]. Mutations in the GJCs building genes lead to oocytes immaturity and cause infertility [69]. The CX43 gene was proposed as a marker of oocyte maturity [70]. The decreased expression of GJA1 was observed in studies on CCs derived from mature ovarian follicles [71], as was downregulation of this gene after IVM of mice cumulus–oocyte complexes [72]. Comparing these data with our results, which indicate downregulation after IVM of swine oocytes, we can assume that GJA1 is a marker of mature oocytes, showing a decrease in expression in relation to immature oocytes.

The second gene that may function as a potential marker of oocytes maturation is MAP3K1 (mitogen-activated protein kinase kinase kinase 1). It was shown that this gene plays a role in follicular atresia, which was associated with inhibition of MAP3K1 expression in porcine GCs, indicating its antiapoptotic effect [73]. Previously, studies on mice GCs showed that MAP3K1 is one of the most important factors influencing fertility, which it does through involvement in the regulation of folliculogenesis and oogenesis [74]. Recent studies on swine oocytes in in vitro culture have also shown downregulation of this gene [75,76]. A decrease in the expression after IVM of the KIT gene (proto-oncogene, tyrosine kinase receptor) was revealed in the current study, and it was recently suggested as a marker of undifferentiated cells of the spermatogenesis pathway in swine testes [77]. Saatcioglu et al. suggested that KIT may function as a factor of oocytes reawakening [78], while studies by Moniruzzaman and Miyano showed that it does not influence the initiation of growth of primary oocytes, but affects their survival [79].

The MMP14 (Matrix metalloproteinase-14) gene is responsible for a wide range of functions associated with angiogenesis, cell migration, and apoptosis [80]. There are various reports on the relationship between MMP14 and ovulation [81,82,83], but the role of this gene in oocytes maturation has not been established. Earlier studies were also concerned with the expression of MMP14 in GCs cells in various species, where an increase in expression was observed after administration of hCG [82] and in the period before ovulation [83]. Due to the wide range of functions performed by MMP14, this gene belongs to “branching morphogenesis of a tube” and “morphogenesis of a branching structure”, and its expression was noted to decrease after IVM.

The next gene that showed a decrease in expression after IVM is inhibit subunit beta A (INHBA). It is believed that it plays an important role in reproduction and development through the coding of activins from TGF-beta superfamily [84,85] and is involved in the secretion of FSH from the pituitary [86], being also a marker of porcine oocyte quality [87]. A study on boars showed that INHBA is significant in testicular development, especially during puberty [88]. Reduced expression of INHBA was demonstrated in studies on swine GCs in vitro culture during follicular atresia analysis [41]. Recently published research on human GCs [86] supports a strong association of this gene with folliculogenesis and ovulation.

Among many genes influencing the morphology and maturity of oocytes, the extracellular component of the matrix, laminin subunit beta 2 (LAMB2), is probably an important factor. It is a noncollagen component of the basement membrane in the ovarian follicle. Studies on bovine ovarian follicles showed that the expression of this gene in GCs cells was different depending on the location of these cells in the follicle, which also resulted in their hormonal activity [89]. LAMB2 downregulation in porcine GCs [41] and porcine oocytes [43] was demonstrated, which correlates with the results obtained during our study. 

The SMAD4 (Mothers against decapentaplegic homolog 4) gene is also strongly associated with ovarian follicle development and GCs function through its association with the TGF-beta signal pathway, moderating apoptosis [90,91]. In addition, it was shown that it is strongly associated with the response of swine GCs to FSH signals [92].

Downregulation of IHH, ITGB1, and CD9 connected with folliculogenesis or oogenesis was revealed after IVM. IHH (Indian hedgehog signaling molecule), whose expression in the mucous membrane of the murine uterus was caused by the addition of progesterone [93], and its expression was documented in swine GCs [25,60], also the ITGB1 (integrin subunit beta 1) gene associated with the embryogenesis processes [94], and the CD9 gene, whose contribution to fertility is not yet clear, although it is known to be important in gametes [95]. CD9 is an important factor associated with oocyte membrane and microvilli, playing an active part in the fusion of oocyte and sperm [96]. 

The expression of the FST gene (follistatin) is observed in many tissues of the body, and its product is also called activin-binding protein. FST plays an important role in the regulation of activin and other members of the TGF-beta superfamily [97,98], which causes effects in the reproductive system, including the formation of female gonads [99]. The expression of FST fluctuates over the course of the reproductive cycle and shows a decrease before ovulation [100]. The downregulation after IVM obtained in our studies may, therefore, indicate that oocytes have reached maturity and may predispose FST to become a marker of maturity of porcine oocytes in vitro.

A decrease in expression of a large number of genes, which until now were not directly or indirectly related to reproductive processes of the maturation of oocytes or folliculogenesis, was revealed in the current study. The genes exhibiting a decrease in expression were involved in the broadly understood angiogenesis and development of the cardiovascular system, although the mechanisms involved in the building of the corpus luteum after ovulation are often based on the construction of vessels and remodeling of the resulting hemorrhagic corpus luteum. VEGFA (vascular endothelial growth factor A), which may have a role in oocyte maturation, [101], CYR61 (cellular communication network factor 1) [102,103], EDNRA (endothelin receptor type A) [104], NEBL (nebulette), the expression of which has been demonstrated in pig GCs [25,60] and is involved in the construction of muscle sarcomeres, and TGFBR3 (transforming growth factor beta receptor 3) were associated with the formation of coronary arteries [105]. Downregulation of genes associated with angiogenesis may be the result of cultures of oocytes themselves, deprived of granulosa cells and other cells involved in corpus luteum formation after ovulation in vivo.

An analogous situation corresponds to the second subgroup of genes, which, in the vast majority, relate to the nervous system, including neurite outgrowth and neural development, and appear here: EGR2 (early growth response 2) [106], CTNNA2 (catenin Alpha 2) [107], RTN4 (reticulon 4) [108], SLITRK3 (SLIT and NTRK like family member 3) [109], ROBO2 (roundabout guidance receptor 2) [110], APP (amyloid beta precursor protein) [111], SEMA5A (semaphorin 5A) [112], and RYK (receptor like tyrosine kinase) [113]. 

The last subgroup of genes that are characterized by many different functions with a wide range of mechanisms, but are usually not related to ovaries. The CDK6 (cyclin dependent kinase 6), ZCCHC11 (also known as TUT4, terminal uridylyl transferase 4), WWTR1 (WW domain containing transcription regulator 1), UBE2B (ubiquitin conjugating enzyme E2 B), NOTCH2 (notch receptor 2), SPTA1 (spectrin Alpha, erythrocytic 1), TPM1 (tropomyosin 1), which may be linked to oocyte maturation [76], and CAPZA2 (capping actin protein of muscle Z-line subunit Alpha 2) genes showed reduced expression. Their relationship with folliculogenesis requires more thorough studies and evidence.

## 5. Conclusions

The present study was based on estimation of CGs distribution in the cytoplasm of oocytes acquiring meiotic competence and determination of their gene expression profile. The above changes are caused by many factors and should be further studied. During the maturation of oocytes, the density of CGs in the cytoplasm decreases. It is also worth noting that the distribution of CGs during oocyte maturation becomes more peripheral and probably occurs in concert with cytoskeleton filaments inside the oocyte. The described ARHGEF2 and MAP1B genes, whose increased expression was observed during this study, are closely related to the development of the cytoskeleton, and may therefore constitute an important new marker for the acquisition of meiotic competence of the oocyte. Among the analyzed ontological groups of genes, the increased expression group only concerned genes from the “cellular component morphogenesis” group, which includes the previously mentioned ARHGEF2 and MAP1B genes. Significant changes in the expression level of the examined genes before and after IVM of porcine oocytes may indicate their potential use as markers of competence acquisition by the oocyte. However, it should be noted that our results may serve as a preliminary study and a reference point for subsequent studies. Additionally, the advantage of the study is the research on a pig model, whose genetic homology to humans is very high. Therefore, the obtained results could be used to improve methods of acquiring oocytes of the best competence of both animals and, after validation in humans, in carrying out in vitro fertilization. 

## Figures and Tables

**Figure 1 genes-11-00815-f001:**
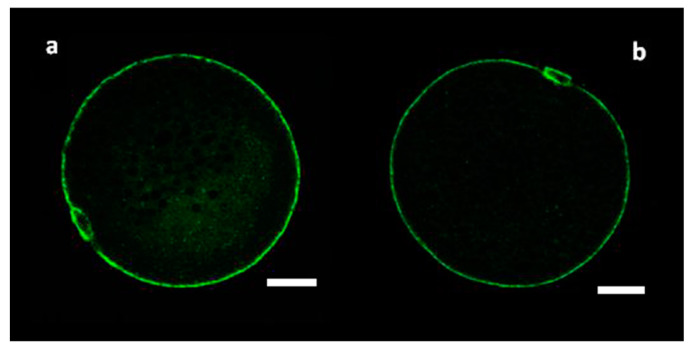
Cortical granule localization before in vitro maturation (IVM). Representative images of porcine oocytes after isolation of oocytes from small (**a**) and medium (**b**) follicles. Oocytes were stained by PNA-labeled FITC (green color—cortical granules). Scale bar represents 20 µm.

**Figure 2 genes-11-00815-f002:**
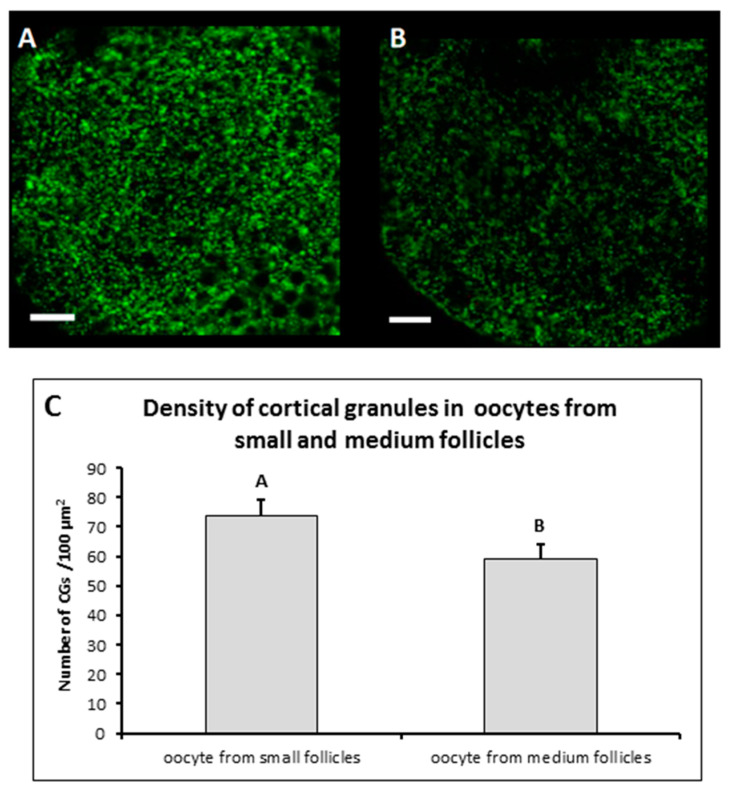
Cortical granule density after IVM. Representative images of porcine oocytes periphery after 44 h maturation in oocytes from small (**A**) and medium (**B**) follicles. Oocytes were stained by PNA-labeled FITC (green color—cortical granules). Scale bar represents 10 µm. The mean (±SD) of cortical granule density was significantly higher in oocytes from small follicles than in oocytes from medium follicles (**C**).

**Figure 3 genes-11-00815-f003:**
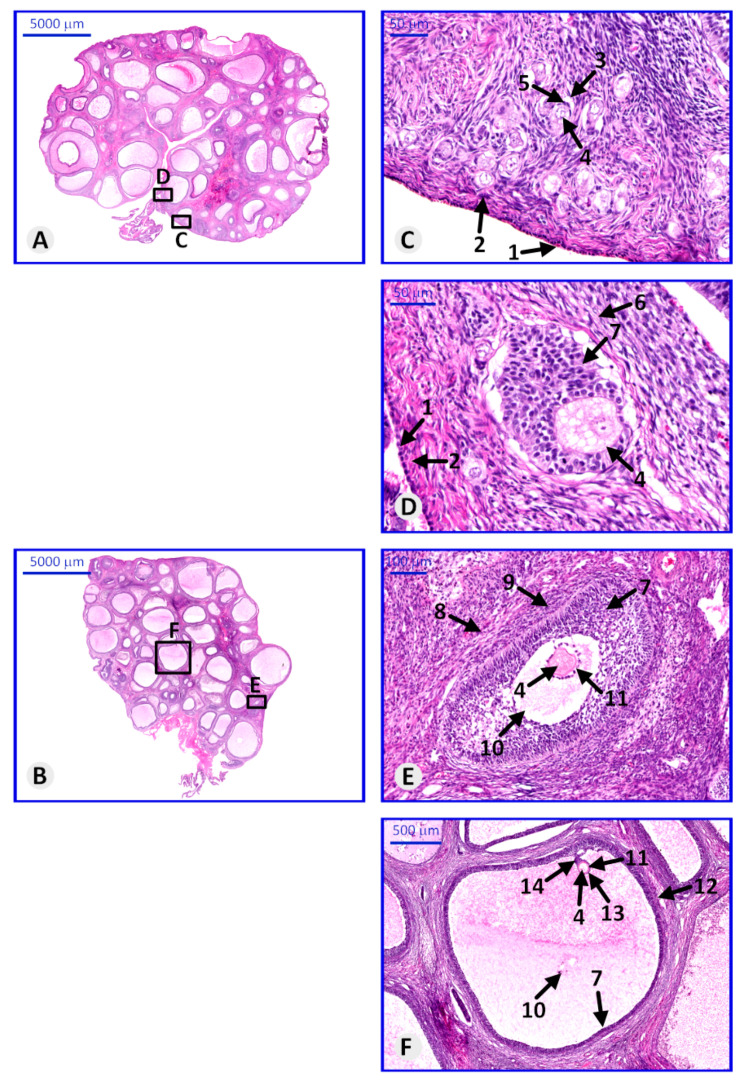
Paraffin section of porcine ovary stained with hematoxylin and eosin (H&E), representing the histological structure of ovary and follicles in particular stages of maturation. (**A**,**B**)—representative ovaries with visible follicles in small magnification; (**C**)—primordial follicles, (**D**)—primary follicle, (**E**)—secondary follicle, (**F**)—Graafian follicle. Arrows: 1—germinal epithelium, 2—tunica albuginea, 3—primordial follicle, 4—oocyte, 5—follicular cells, 6—primary oocyte, 7—granulosa cells, 8—secondary follicle, 9—tunica interna and externa, 10—antrum, 11—zona pellucida, 12—Graafian (mature) follicle, 13—corona radiate, 14—cumulus oophorum.

**Figure 4 genes-11-00815-f004:**
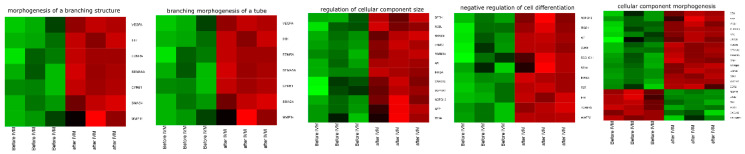
Heat map representations of differentially expressed genes belonging to “cellular component morphogenesis”, “negative regulation of cell differentiation”, “morphogenesis of a branching structure”, “regulation of cellular component size”, and “branching morphogenesis of a tube” GO BP terms. Arbitrary signal intensity acquired from microarray analysis is represented by colors (green, higher; red, lower expression). Log2 signal intensity values for any single gene were resized to Row Z-Score scale (from −2, the lowest expression to +2, the highest expression for single gene).

**Figure 5 genes-11-00815-f005:**
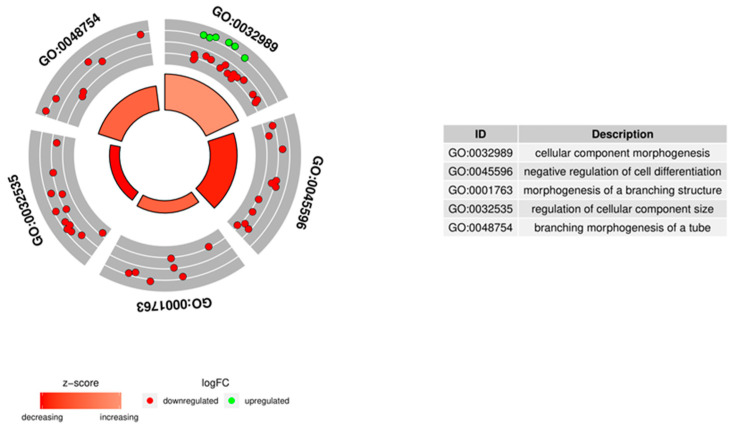
Circle plot showing the differently expressed genes and Z-scores of the “cellular component morphogenesis”, “negative regulation of cell differentiation”, “morphogenesis of a branching structure”, “regulation of cellular component size”, and “branching morphogenesis of a tube” Gene Ontology Biological Process (GO BP) terms. The outer circle shows a scatter plot for each term of the fold change of the assigned genes. Green circles represent upregulation and red ones represent downregulation. The inner circle shows the Z-score of each GO BP term. The width of each bar corresponds to the number of genes within GO BP term and the color corresponds to the Z-score.

**Figure 6 genes-11-00815-f006:**
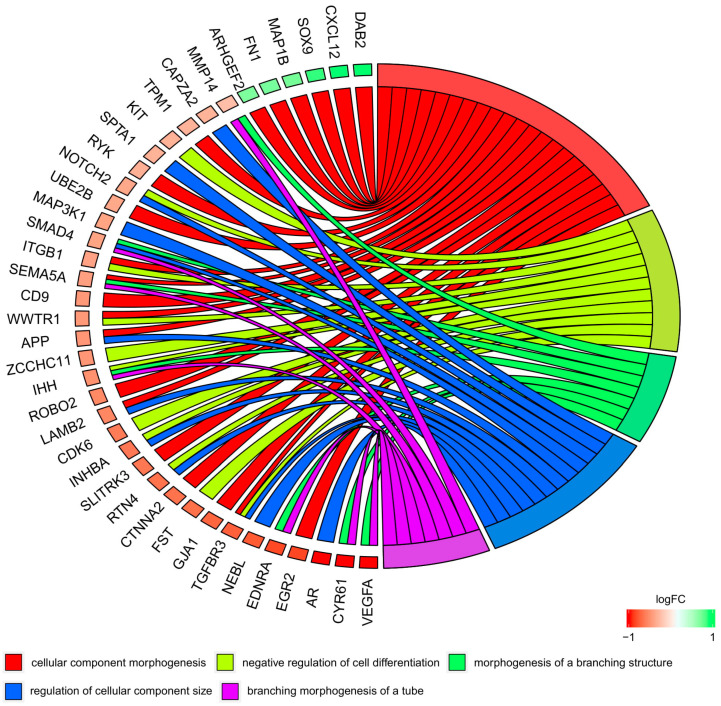
The representation of the mutual relationship between differently expressed genes that belong to the “cellular component morphogenesis”, “negative regulation of cell differentiation”, “morphogenesis of a branching structure”, “regulation of cellular component size”, and “branching morphogenesis of a tube” GO BP terms. The ribbons indicate the genes belonging to specific categories. The middle circle represents logarithm from fold change (LogFC). The genes were sorted by logFC by differences in gene expression. The color of the each LogFC bar corresponds to the LogFC value.

**Figure 7 genes-11-00815-f007:**
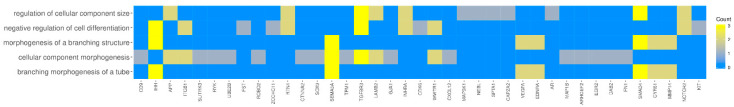
Heatmap showing the gene occurrence between differently expressed genes that belong to the “cellular component morphogenesis”, “negative regulation of cell differentiation”, “morphogenesis of a branching structure”, “regulation of cellular component size”, and “branching morphogenesis of a tube” GO BP terms. The yellow color is associated with gene occurrence in the GO Term. The intensity of the color corresponds to the amount of GO BP terms that each gene belongs to.

**Figure 8 genes-11-00815-f008:**
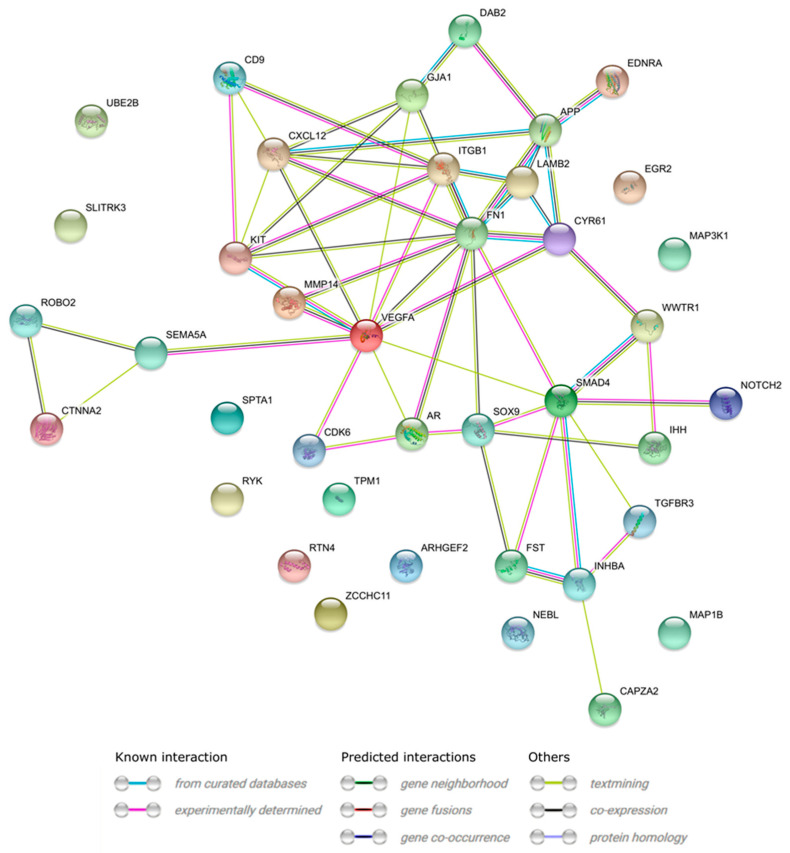
STRING-generated interaction network between genes that belong to the “cellular component morphogenesis”, “negative regulation of cell differentiation”, “morphogenesis of a branching structure”, “regulation of cellular component size”, and “branching morphogenesis of a tube” GO BP terms. The intensity of the edges reflects the strength of interaction score.

**Figure 9 genes-11-00815-f009:**
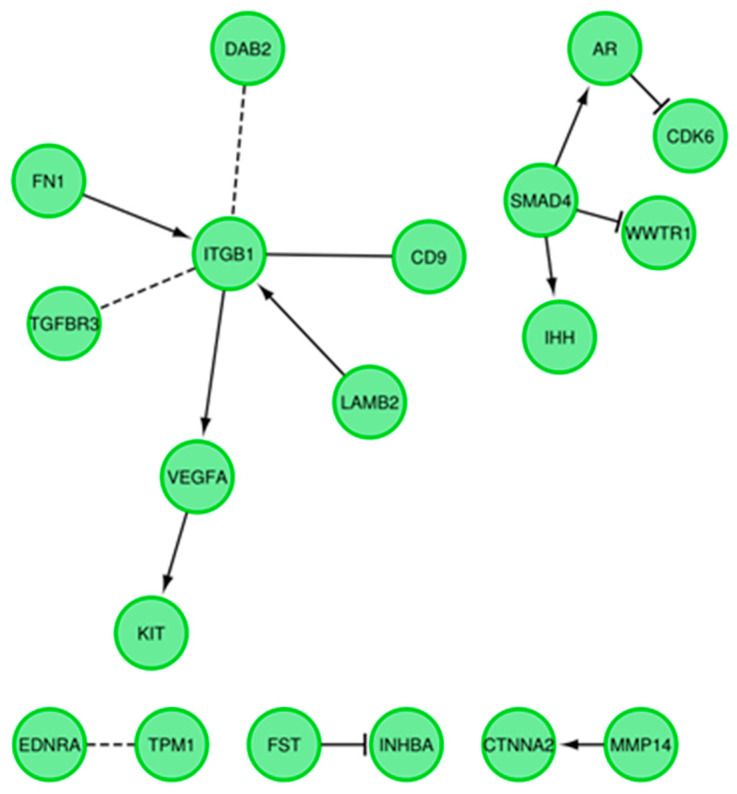
Functional interaction (FI) between differently expressed genes that belong to “cellular component morphogenesis”, “negative regulation of cell differentiation”, “morphogenesis of a branching structure”, “regulation of cellular component size”, and “branching morphogenesis of a tube” GO BP terms. In following figure “->” stands for activating/catalyzing, “-|” for inhibition, “-” for FIs extracted from complexes or inputs, and “---” for predicted FIs.

**Figure 10 genes-11-00815-f010:**
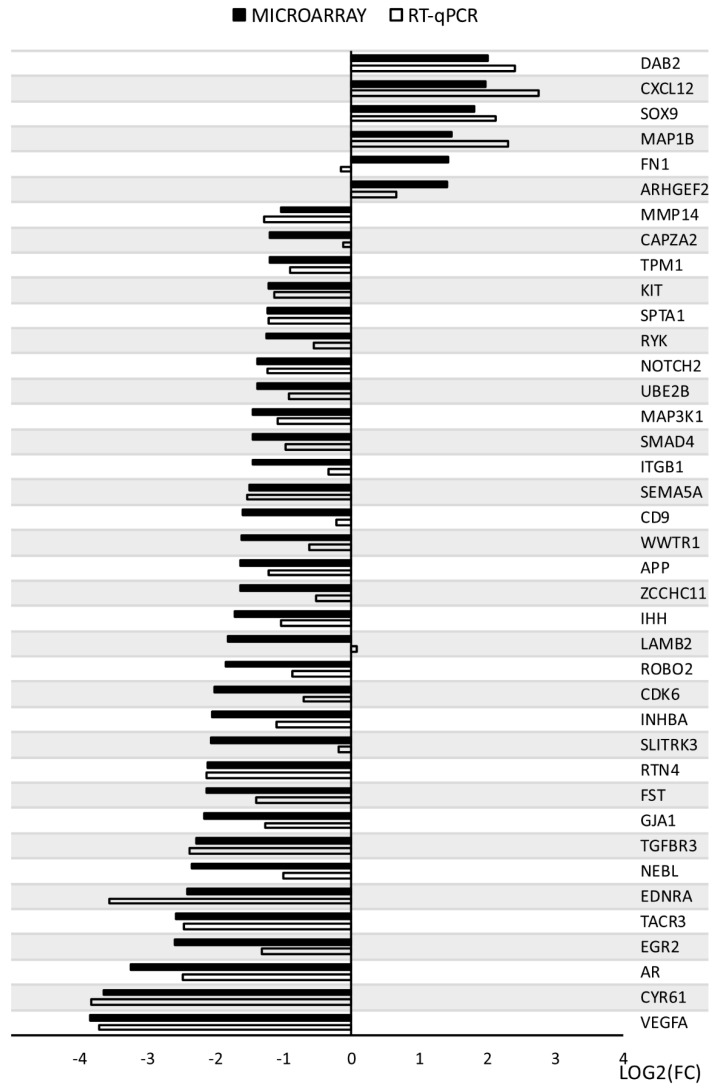
RT-qPCR—microarray validation.

**Table 1 genes-11-00815-t001:** Cortical granule distribution in porcine germinal vesicle stage (GV) oocytes collected from different-sized follicles.

Follicle Size	Oocytes	
	No. of Oocytes Examined	Index of Cortical Granules Distribution I_cgd_ * (Mean ± SD)
Small	134	1.48 ^a^ ± 0.31
Medium	51	2.22 ^b^ ± 0.49

Values with different superscript were significantly different (*p* ˂ 0.01). ***** I_cgd_ = average signal intensity (FITC) in peripheral part of oocyte/average signal intensity (FITC) in central part of oocyte. ^a^—small follicles, ^b^—medium follicles.

**Table 2 genes-11-00815-t002:** Gene symbols, ratio of fold change in expression, corrected *p* values, and LogFC of studied genes.

Gene	Gene ID	Fold Change	*p* Value
VEGFA	397157	0.069689389	0.001912689
CYR61	100153791	0.080657036	0.0000754
AR	397582	0.1059863	0.000138367
EGR2	100038004	0.165503832	0.007949861
EDNRA	397457	0.166939028	0.00185422
NEBL	100522395	0.187213468	0.005937926
TGFBR3	397512	0.196522244	0.000405979
GJA1	100518636	0.206907347	0.000107676
FST	445002	0.224696558	0.000364693
CTNNA2	100525337	0.229925246	0.000512181
RTN4	100170118	0.23137773	0.027495815
SLITRK3	106504067	0.239141911	0.004260951
INHBA	397093	0.241259771	0.000148036
CDK6	100518921	0.248001413	0.006042481
LAMB2	101101688	0.278857947	0.000187911
ROBO2	100517681	0.283748864	0.001183495
IHH	397174	0.304995843	0.000551261
ZCCHC11	100516979	0.3216223	0.019809962
APP	397663	0.324138605	0.005602323
WWTR1	100522573	0.327202092	0.000254025
CD9	397067	0.329283105	0.006332387
SEMA5A	100737194	0.353391715	0.001092396
ITGB1	397019	0.366233017	0.003705215
SMAD4	397142	0.367802201	0.001238681
MAP3K1	396617	0.36876538	0.024748462
UBE2B	100513527	0.382779667	0.041104659
NOTCH2	100153369	0.3848262	0.002523723
RYK	100523513	0.421903006	0.00439989
SPTA1	100152068	0.427042121	0.002476662
KIT	396810	0.430444215	0.00255635
TPM1	100037999	0.433963109	0.001632742
CAPZA2	100037958	0.434375309	0.017755958
MMP14	397471	0.488721147	0.038060423
ARHGEF2	100145887	2.656199858	0.006431804
FN1	397620	2.679580656	0.001210303
MAP1B	100519062	2.788554105	0.00105385
SOX9	396840	3.503340174	0.000620008
CXCL12	494460	3.934705791	0.003163632
DAB2	100519746	4.008208178	0.001912689

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
