# Peer review of "Cortical Granule Distribution and Expression Pattern of Genes Regulating Cellular Component Size, Morphogenesis, and Potential to Differentiation are Related to Oocyte Developmental Competence and Maturational Capacity In Vivo and In Vitro"

_genes, 2020, doi:10.3390/genes11070815_

Round 1
Reviewer 1 Report
The objective of the study was to understand the expression of genes related to morphogenesis before and after in vitro maturation (IVM) of swine oocytes to better define gene pathways regarding in maturation and cortical reaction. Furthermore, an aim was to better understand distribution and density of cortical grains (CGs) in porcine oocytes in regard to mitotic competency for IVM porcine oocytes. Before the study accepted for publication the following points should be addressed.
1- The introduction section is short and requires more information and extension of background
2- Statistical analysis is not clear particularly on the part of distribution of cortical granules
3- Why the authors did not perform IVF experiments?
4- What is the significance to use BCB staining in this study?
5- Why the authors used ovaries collected from adult sows and not from prepubertal gilts?
6- Discussion part is too long and included many unnecessary information, particularly line 375 to 401 (this could be in introduction)
7- Extensive uses of abbreviations which gives unclarity and confusion for the readers
8- Many spellings and grammatic mistakes
9- Line 43, use word for after processed
10- Line 55, change polispermia to polyspermia
11- Line 76 change estral cycle to estrous cycle
12- Line 87, correct cortical grains and remove CGs. This is common mistakes all over the manuscript, the authors tended to add the abbreviations and the full words more than one time. Please, mention the full name and then the abbreviation only once in the beginning and only repeat the abbreviation later
13- Line 84, please identify the meaning of adequate morphological status
14- Line 97-101, why the authors used 2 methods to collect the oocytes. Although they mixed all pooled COCs together?
15- Line 113-114, this sentence should be the first one in materials and methods.
16- Line 114, used oocyte-cumulus complexes, although, before this part the authors used cumulus-oocyte complexes. Please, be consistence all over the manuscript. Using cumulus-oocyte complexes (COCs) is more acceptable.
17- Line 357, change A,B to a,b as in the pictures
18- Line 378, change share to rate
19- Line 387, what this ward means (inter alia)?
20- Line 405, correct this phrase “There appears”
21- Line 415, re-write this An emphasis of the was the
22- Line 448, cortical granules (CGs) you already mentioned the full name before
23- Line 464, what is this abbreviation (OCs) mean in swine oocytes (OCs)
Reviewer 2 Report
The Authors investigate the differences of cortical granules (GCs) localization in immature oocytes isolated from medium and small porcine follicles before and after their in vitro maturation. Moreover, they analyze with microarray, by using an Affimetrix panel, the gene expression profile of immature and mature oocytes and confirm data by Real Time PCR and try to correlate the different localization of GCs with the deregulated expression of some genes. However, in my opinion the two experimental parts (cellular and molecular) of the work are disconnected and the Authors should make an effort to better integrate the two sides. In fact, although the localization of the GCs seems to correlate with both size of the follicles and their maturation, there is no convincing evidence that the oocyte gene expression profiles correlate with the differences in the GCs density and segregation.
Major criticism
- The ‘Materials and methods’ paragraph is too long and fragmented. The Authors should merge and shorten some sub-paragraphs (e.g. 2.1 and 2.2; 2.3, 2.4 and 2.5; 2.6, 2.7 and 2.8).
- The sequence of ‘Results’ is reversed with respect to ‘Materials and methods’. The Authors should maintain the same consequentiality since microarray analyses were performed after the selection and subsequent in vitro maturation of the oocytes. Therefore the sequence of the figures should also be changed.
- Figure 9 is quite confusing, since the peripheral reactivity of GCs is more evident in oocyte from small follicle (a) than in those from medium follicle (b), although in fig. 9a is represented also the central reactivity. The evidence is that in fig. 9a the oocyte exhibits greater reactivity overall.
- The Authors should also add the images of the oocytes examined with the nuclear dye in microscopy before and after in vitro maturation, to highlight the quality of the oocytes used in subsequent experiments.
- It is not clear whether the microarray experiments were conducted on oocytes from small or medium follicles or if it was done on a pool of these before and after in vitro maturation. However, if the latter statement is true, it would have been interesting to investigate possible differences in in vitro maturation of oocytes from follicles of different sizes. The authors should clarify this point in the ‘Results’ and comment in the ‘Discussion’.
Minor criticism
- Page 2, line 87: eliminate ‘cortical grains’ since the abbreviation GCs is reported at line 61 in the same page.
- Page 2, line 87: substitute ‘meitotic’ with ‘meiotic’.
Round 2
Reviewer 2 Report
The Authors answered to the criticisms completely and convincingly. Moreover, they modified the manuscript following the Reviewers' suggestions. I believe that it is suitable for publication in the current form.
